# RETRACTED: Fluorescent and Phosphorescent Nitrogen-Containing Heterocycles and Crown Ethers: Biological and Pharmaceutical Applications

**DOI:** 10.3390/molecules27196631

**Published:** 2022-10-06

**Authors:** Faiz Ullah, Sami Ullah, Muhammad Farhan Ali Khan, Muhammad Mustaqeem, Rizwan Nasir Paracha, Muhammad Fayyaz ur Rehman, Fariha Kanwal, Syed Shams ul Hassan, Simona Bungau

**Affiliations:** 1Department of Chemistry, Quaid I Azam University, Islamabad 45320, Pakistan; 2Department of Zoology, Government College University, Faisalabad 38000, Pakistan; samibhatti925@gmail.com; 3Faculty of Pharmacy, Capital University of Science and Technology, Islamabad Expressway, Islamabad 44000, Pakistan; farhanali@bs.qau.edu.pk; 4Institute of Chemistry, University of Sargodha, Sargodha 40100, Pakistan; muhammad.mustaqeem@uos.edu.pk (M.M.); fayyaz9@gmail.com (M.F.u.R.); 5Department of Chemistry, Sub Campus, University of Sargodha, Bhakkar 30000, Pakistan; rizwan.nasir@uos.edu.pk; 6School of Biomedical Engineering, Shanghai Jiao Tong University, 1954 Hua Shan Road, Shanghai 200030, China; farihakaanwal@gmail.com; 7Shanghai Key Laboratory for Molecular Engineering of Chiral Drugs, School of Pharmacy, Shanghai Jiao Tong University, Shanghai 200240, China; 8Department of Natural Product Chemistry, School of Pharmacy, Shanghai Jiao Tong University, Shanghai 200240, China; 9Department of Pharmacy, Faculty of Medicine and Pharmacy, University of Oradea, 410028 Oradea, Romania

**Keywords:** fluorescence, heterocyclic compounds, antitumor, antifungal, anti-microbial

## Abstract

Fluorescent molecules absorb photons of specific wavelengths and emit a longer wavelength photon within nanoseconds. Recently, fluorescent materials have been widely used in the life and material sciences. Fluorescently labelled heterocyclic compounds are useful in bioanalytical applications, including in vivo imaging, high throughput screening, diagnostics, and light-emitting diodes. These compounds have various therapeutic properties, including antifungal, antitumor, antimalarial, anti-inflammatory, and analgesic activities. Different neutral fluorescent markers containing nitrogen heterocycles (quinolones, azafluoranthenes, pyrazoloquinolines, etc.) have several electrochemical, biological, and nonlinear optic applications. Photodynamic therapy (PDT), which destroys tumors and keeps normal tissues safe, works in the presence of molecular oxygen with light and a photosensitizing drugs (dye) to obtain a therapeutic effect. These compounds can potentially be effective templates for producing devices used in biological research. Blending crown compounds with fluorescent residues to create sensors has been frequently investigated. Florescent heterocyclic compounds (crown ether) increase metal solubility in non-aqueous fluids, broadening the application window. Fluorescent supramolecular polymers have widespread use in fluorescent materials, fluorescence probing, data storage, bio-imaging, drug administration, reproduction, biocatalysis, and cancer treatment. The employment of fluorophores, including organic chromophores and crown ethers, which have high selectivity, sensitivity, and stability constants, opens up new avenues for research. Fluorescent organic compounds are gaining importance in the biological world daily because of their diverse functionality with remarkable structural features and positive properties in the fields of medicine, photochemistry, and spectroscopy.

## 1. Introduction

Molecular luminescence approaches include phosphorescence and fluorescence. A photon is absorbed by an analyte molecule, which stimulates a species. The emission spectrum can be used for quantitative and qualitative studies [1,2]. Because of their potential various functional applications, luminous materials have received significant attention lately [3]. They have been widely employed, including in the food, pharmaceutical, optical, and textile sectors [4,5]. Conventionally, inorganic emitting materials were commonly used; however, organic luminescent materials with brilliant emission have largely replaced them due to their wide range of uses, including emergency lighting, low cost, environmental friendliness, long-term solutions, anti-counterfeiting displays, and in food, cosmetics, polymers, bioactive molecules, and biochemistry [6,7]. Furthermore, in today’s research, the design of novel luminous hybrid organic–nonorganic materials is critical [8,9]. The combination of phosphorescent dye as a sensitizer co-doped with a fluorescence emitter has made progress in developing luminescent materials for organic light-emitting diodes (OLEDs) in recent years [10,11].

Fluorescence quenching or fluorescence enhancement is employed as an analytical technique [12,13]. The fluorescent labelling of the host molecule complex provides a useful tool for detecting the analyte’s attachment to the host molecule [14,15,16]. For example, protein labelling using small molecule-based fluorescent probes is used in various biological experiments and is a valuable technique for determining the expression level and localization of a protein of interest in living cells [17].

Due to particular biological activity, crown ether’s derivate and *N*-containing heterocyclic chemicals and their derivatives have been widely employed in agronomy and medicine [18]. Similar organic chemicals are of interest in pharmacology as effective tissue oxygenators and antidepressants, as well as in biotechnology; these compounds are employed for macromolecule binding [19,20,21,22,23]. *N*-containing heterocyclic compounds and macrocycle derivative crown ethers have remarkable photochemical, catalytic, and luminescent capabilities, indicating that they might be used to diagnose and cure various ailments. A few of their applications include photodynamic treatment and antimicrobial/antiparasitic activities against human pathogens and malarial parasites. Employment of fluorophores, including organic chromophores and crown ethers, with high selectivity, sensitivity, and stability constants while detecting tumor cells opens up new avenues for cancer research [24,25].

Macrocyclic molecules, for example, crown ether, have been used in a wide range of chemical processes, including selective metal complexing agents and photo-induced electron transfer bio-mimetic research [26,27,28,29]. In contrast to the extensive coordination chemistry, little is known about crown ether coordination compounds’ photoluminescence (PL). Crown ethers substituted with particular fluorescent dyes were the most commonly reported for PL. The use of such dye-substituted systems in sensing and analytical chemistry to detect the presence of particular metal cations was intensively investigated [30,31,32].

A smart fluorescent probe with a crown ether moiety might be constructed as a sensor for metal anions, ions, and other biomolecules and then used to monitor biological processes in vivo [33]. The solvent effects of a crown ether complex containing a fluorescent anthracene unit are exceptional [34].

Supramolecular chemistry, inspired by nature’s vast array of assemblies, has garnered significant attention in recent decades due to its diverse supra-structures, which consist of micelles, vesicles, and fibers, as well as its wide-ranging applications in sensors, drug delivery, luminescent materials, and bioimaging [35,36,37,38].

Fluorescence characteristics of *N*-containing heterocyclic compounds have recently received considerable interest. For example, fluorescent compounds known as quinolines have attracted the attention of scientists because of their use in high-tech applications [39]. Similarly, derivatives of the pyrazoloquinoline (PQ) family and quinoline are an example of fluorescent substances that may be of interest for several applications, including their use as oxidant scavengers and growth promoters [40,41]. These have also been found naturally in a wide range of foods and appear to be easily absorbed. More recently, heterocyclic azo compounds such as benzothiazole, pyrazole, and thiazole have been employed for electrochemical, biological, and nonlinear optics applications and structure–activity relationships for drug designing [SAR] [42,43,44]. Thiophene and thienopyrimidine derivatives have fluorescence features and are more efficient than other aromatic chemicals for anti-avian influenza virus (H5N1) action. Porphyrins are N-heterocyclic chemicals present in a wide variety of biological systems. Metalloporphyrins contain solely -pyrrolic substituents in biological systems and appear attached to proteins, creating supramolecular structures such as haemoglobin, myoglobin, cytochromes, catalases, and peroxidases, as well as chlorophylls and bacteriochlorophylls in reduced forms [45].

Within the constraints of this review, it is not feasible to address the fluorescence characteristics of all compounds of interest in biochemistry and medicine. However, crown ether and *N*-containing heterocyclic compounds that show fluorescence capabilities are chosen for this section to demonstrate their biological and pharmaceutical applications in daily life.

## 2. Applications of Heterocyclic Compounds

### 2.1. Anti-Mycobacterial Activity

Different symptoms such as respiratory issues, long-term coughs, and tuberculosis are treated by various plants in African and Asian countries. Many anti-tubercular drugs, with toxicity and side effects, are still used to treat tuberculosis. For treating *M. tuberculosis*, the synthesis of azo compounds was monitored and showed anti-tubercular activity. Maximum activity was shown by compounds 5-methyl-2-(5-methylbenzo[*d*]thiazol-2-yl)-4-(*p*-tolyldiazenyl)-1*H*-pyrazol-3(2*H*)-one **(1a)** and 5-methyl-2-(5-methylbenzo[*d*]thiazol-2-yl)-4-(*m*-tolyldiazenyl)-1*H*-pyrazol-3(2*H*)-one **(1b)** when compared to the copounds 4-((4-chlorophenyl)diazenyl)-5-methyl-2-(5-methylbenzo[*d*]thiazol-2-yl)-1*H*-pyrazol-3(2*H*)-one **(1c)** and 4-((4-bromophenyl)diazenyl)-5-methyl-2-(5-methylbenzo[*d*]-thiazol-2-yl)-1*H*-pyrazol-3(2*H*)-one **(1d)** shown below in Figure 1, correspondingly. A previous study shows that the presence of a side chain to an azo dye along with a phenyl group substituent and a significantly enhanced electron-donating group ultimately decreased the growth of bacteria [5].

### 2.2. Anticancer Activity

The photochemistry and the anti-tuberculosis activity of the in vitro azo compounds discussed above yielded good results, so their anticancer activity was also studied. An MTT test was performed for cell proliferation, and for this reason, different human cancer cell lines were used, such as chronic myeloid leukaemia (K562), lung carcinoma (A549), colon (HCT116), and T-lymphocyte (Jurkat) cell lines. Table 1 shows their anticancer activity results. Data revealed that K562, Jurkat, and A549cell lines containing various synthesized azo compounds displayed fair in vitro results (IC_50_ > 50). However, on the other hand, in comparison with other human cell lines, the HCT116 cell line showed relatively good activity in the presence of various compounds [46].

### 2.3. Therapeutic and Biological Applications

Various applications, such as anti-inflammatory, antibacterial, analgesic, antiviral, antipyretic, and anti-convulsant activities, belonged to 3-aminopyrroles derivatives, which are considered an essential family of compounds [47]. Thiophene compounds also play a significant role as agrochemicals [48,49], anti-avian influenza virus (H5N1), anti-tubercular, anti-breast cancer agents, AMPK activators, HIV, and multi-target kinase inhibitors [50].

The majority of roles, including serving as precursors for different biological molecules or connecting to various sulphur and nitrogen heterocycles, are imparted by some structural units combined to form a 2-aminothiophene product. Apart from this, UV-visible absorption and fluorescence of these compounds make them important for biological purposes. Thiophene derivatives can be used explicitly as valuable fluorescent dyes in confocal microscopy for bio-imaging [51].

### 2.4. Antiparasitic Activity of Metalloporphyrins and Their Role as Potentiometric Biosensors

Metalloporphyrins, known for their β-pyrrolic substitution, are important in forming useful supramolecules such as cytochromes, haemoglobin, peroxidases, myoglobin, and catalases [52,53]. The main reason porphyrins are gaining importance in the biological world day by day is their diverse functionality along with their remarkable structural features and positive properties in the field of photochemistry and spectroscopy. The use of metalloporphyrins as potentiometric sensors is common among all other functions—for example, Mn(III)-porphyrin derivatives are being used in the chloride ion measurement in samples of human serum [54].

The increase in antiparasitic activity of porphyrins is related to the presence of electrically charged substituents on these compounds. An ultimate decrease in the oxidative damage to the mosquitoes’ larvae of genera *Culex, Aedes* [55,56], and *Anopheles* [57]*,* while of adult flies of *Ceratitis capitates*, *Bactrocera oleae* species, and *Stomoxys calcitrans* [54,58] can be observed by porphyrin-based drugs. Photosensitization makes hematoporphyrin IX a powerful eco-friendly drug.

### 2.5. Antioxidant Activity

Disordered physiological processes such as neurodegenerative disorders are studied by reactive nitrogen and oxygen species or heterocyclic compounds [59]. Neuroprotection involves an option of antioxidant therapy, so antioxidants can be described as compounds capable of searching for free radicals. Dicsussing specific fluorescent heterocycles shown in Figure 2 such as (3s,5s,7s)-*N*-(2,4-dinitrophenyl)adamantan-1-amine (**2a**), *N*-((3s,5s,7s)-adamantan-1-yl)-6-(dimethylamino)-naphthalene-2-sulfonamide (**2b**), 2-(adamantan-1-yl)-2*H*-isoindole-1-carbonitrile (**2c**), provide us a guide to the pharmacological industry as they are of great interest as antioxidant agents [60].

### 2.6. Neuroprotective Agents

In addition to antioxidant properties, fluorescent heterocyclic aminoadamantane compounds exhibit neuroprotection and can serve as active compounds in search of potential therapeutics. Aside from their medical significance, many of these compounds are yet to be studied for their toxicity in humans. With further pharmacological studies and the development of fluorescent displacement, aminoadamantane derivatives can be used for radio ligand binding and neurodegenerative process explorations. In the biological and pharmacological industries, the function of these fluorescent heterocyclic compounds as neuro-protective drugs should be further investigated as they have encouraging physical and chemical properties and can also be used as fluorescent ligands [60].

### 2.7. Bioorganic Activity of 1,4-Dihydropyridines

The 1,4-dihydropyridines compounds are highly important as they are considered beneficial for bioorganic, synthetic, and therapeutic chemistry [61]. In biological systems, these compounds show an interesting reduction in strained ring systems such as epoxides, conjugated olefins, and carbonyls, etc., and also in unsaturated functional groups. Their unique ability involves coenzyme reduced nicotinamide adenine dinucleotide (NADH). It is said that nifedipine, belonging to a class of 1,4-dihydropyridine, shows photo toxicity. The oxidation and photo-oxidation processes of 1,4-dihydropyridines are being investigated due to their large demand and interest [62].

### 2.8. Antihypertensive and Antibacterial Activity

In this work, an antihypertensive agent 2-(2,6-dichlororbenzylidenehydrazino)-1,4,5,6-tetrahydropyrimidine hydrochloride (**3**) (OT-24) was synthesized as the anti-isomer (*E*-isomer) by the experimentation of the nuclear Overhauser effect (NOE) and, by a process of irradiation with ultraviolet light in an aqueous or methanolic solution, it was instantly converted to its syn-isomer (*Z*-isomer). Not long ago, the compound (*Z*)-2-(2-(2,6-dichlorobenzylidene)hydrazinyl)pyrimidine (**4**) and its related 2-benzylidenehydrazinopyrimidine derivatives exhibited remarkable antibacterial activity as shown below in Figure 3 [63].

### 2.9. Anti-Microbial, Antifungal and Antitumor Activities of Metal N-Heterocyclic Carbine Complexes

The ionic silver complexes such as AgNO_3_ attracted great attention due to their increased stability, which was considered favorable for antimicrobial activity. Then, silver *N*-Heterocyclic Carbene (NHC) complexes were encapsulated, by electro-spinning, into polymers. This change led to an increase in their antifungal and bacteriostatic potential. Additionally, the anticancer activity of metal–NHC complexes has been reviewed and reported recently. The complexes showed cytotoxicity whenever a metal was bound to an NHC ligand. Cisplatin, in particular, was outshone when metals such as silver, copper, palladium, and gold formed complexes and displayed significant antitumor activities as in Figure 4 given below—compounds such as bis(1-benzyl-3-(*tert*-butyl)-2,3-dihydro-1*H*-imidazol-2-yl)palladium(IV) chloride (**5**), (1,3-dimesityl-2,3-dihydro-1*H*-imidazol-2-yl)copper(II) chloride (**6**) and (1,3-dipropyl-2,3-dihydro-1*H*-imidazol-2-yl)silver(II) chloride (**7**), respectively [64].

### 2.10. Anti-Malarial, Anti-HIV, and Antibacterial Activities of Carbazoles

Collins and co-workers designed a method for the synthesis of a carbazole heterocycles family. Currently, this work is being extended by the same group using a different technique of photochemistry and two-step continuous-flow processes to achieve a more complicated carbazole structure [65]. A diverse range of carbazoles can be made using photochemical decomposition of azides. These carbazoles, when transformed into family alkaloid clausine C, are immensely important from a biological perspective as anti-HIV, antibacterial, and antimalarial agents, while carprofen (2-(9*H*-fluoren-2-yl) propanoic acid) is important as an anti-inflammatory agent, as shown below in Figure 5 [66].

### 2.11. 1,2,4-Oxa-diazoles Activity as Peptidomimetics and Bioisosteres

Because of their pharmaceutical roles, 1,2,4-oxadiazole derivatives are gaining importance. The photo-reactivity of particular 1,2,4-oxadiazoles significantly depends upon the perfluoroalkyl group [67]. Among various fluorinated five-membered heterocycles, a number of properties exhibited by 1,2,4-oxadiazoles were known to be dependent on a functional group present at C(3) position. Due to their having great importance in the pharmaceutical industry, 1,2,4-oxadiazoles have been used as bioisosteres for esters and amides and as peptidomimetics, while 3-amino derivatives of these compounds were shown to be powerful and effective muscarinic agonists [68].

### 2.12. Anti-Microbial Activity of 2-Chloro-5-methylpyridine-3-olefin Derivatives

In modern molecular photobiology and photochemistry, photochemical *E/Z* isomerization is greatly valued. To synthesize 2-chloro-5-methylpyridine-3-olefin derivatives (**8a**–**e**), 2-chloro-5-methylnicotinaldehyde can be used, and their *E*→*Z* (**9a**–**e**) isomers were studied. It was seen that the triplet excited state showed better *E* (*trans*) →*Z* (*cis*) isomerization compared to the singlet excited state. As pointed out by fluorescence studies, these isomerizations involved a polar singlet excited state or transfer of charge. As shown in Figure 6, 2-chloro-5-methylpyridine-3-olefin derivatives (**8a**–**e**) and their *E→Z* (**9a**–**e**) isomeric compounds were monitored, and they showed moderate anti-microbial activity [69].

### 2.13. Antitumor Activity

A new class of amidino- and cyano-substituted naphtha [2,1-β] furans and naphtha [2,1-β] thiophenes were developed. These compounds were found to exhibit antitumor activity, served as DNA intercalators, and, in addition, were somehow linked to thienobenzofurans, naphthofurans, benzothiophene, and naphthothiophenes [70].

### 2.14. Antioxidant Activity of Halogenated β-Carbolines

Under photo-induced oxidative stress, β-carbolines (βCs) were considered good structures to show antioxidant activity. The antioxidant properties were further explored to understand the different biological functions of β-carbolines [71].

### 2.15. Antioxidants and Various Other Important Roles

Flavonoids are associated with stable radicals’ formation and instant oxidation and are known to protect from damage caused by free radicals, and they have a polyphenolic nature with antioxidant activity. The damage caused by free radicals was caused by various metabolic processes and singlet oxygen produced by the photolytic processes in living organisms [72]. Flavonoids also hold a grip on different biological impacts; when ultraviolet β-radiations cause damage, flavonoids are used to protect against them. These compounds also reduce cholesterol absorption and improve blood flow [73] (Table 2). The molecules that could not be accessed by conventional chemistry were now achieved by photochemical transformations and the synthesis of flavonoids. Another milestone achieved in this class of compounds was better photochemistry and photostability of flavonoids, which resulted in their increased use as food additives for health purposes and as important constituents of black tea, adhesives, and red wine on the commercial scale [74].

### 2.16. Photodynamic Therapy of Phthalocyanines

Long ago, an alternative and useful therapy to treat various diseases involved exposing dyes to visible light to inactivate the photodynamic activity of biological systems. Photodynamic therapy (PDT), which destroys tumors and keeps normal tissues safe, works in the presence of molecular oxygen with light and a photosensitizing drug (dye) to obtain a therapeutic effect. Phthalocyanines (Pcs), a class of photosensitizers, are heterocyclic compounds that form chelate complexes with metal cations and consist of nitrogen atoms being used as bridges to link four benzoindole nuclei. Phthalocyanines have been successfully used by incorporating them into liposome membranes and in various other drug delivery systems, including cyclodextrins and oil emulsions systems [75].

### 2.17. Drug Activity and DNA Targeting Activity of Coumarins and Phenanthridines

By incorporating a suitable functional group in phenanthridine moiety, the role of photo-responsive chromophores was easily determined by coumarin and phenanthridine-fused scaffold, which were believed to have a significant impact on the development of organic molecules, specifically on novel coumarin and phenanthridines. This belief was the basis of intrinsic fluorescence properties of coumarin and redox- and light-sensitive properties acquired by phenanthridine derivatives. Coumarins and phenanthridines come under the two major divisions of heterocycles, having a wide range of applications in various fields such as drugs [76], dyes, and DNA targeting agents [77,78].

### 2.18. Applications of Fluorescent Quinolones, Quinolines and Their Derivatives

The importance of quinolines is increasing due to their renowned fluorescent compounds and their use as fluorescence probes in chemosensors [79]. Some compounds show eminent fluorescent properties such as substituted 4-trifluoromethylquinolones (**11a**), 4-cyanoquinolones (**11b**), and 3,4-dicyanoquinolones (**11c**) which are derived from quinolin-2-one (**10a**) and 4-hydroxyquinolin-2-ones (**10b**) as shown below in Figure 7, respectively [80].

In addition, compounds in Figure 8, **12a**–**f,** were also found to exhibit fluorescence and used as excellent fluorescent probes for exposure to bacteria [81], tumor cells [93], or cysteine present in living cells.

Along with the above-mentioned applications, quinoline derivatives were also used to detect metal ions as fluorescent probes. Protein detection was considered an essential function performed by two compounds (**13a**–**b**) of 4-hydroxyquinolin-2-one dyes (Figure 9), which contained 4-diethylamino-2-hydroxyphenyl substituents and displayed high emission with bright fluorescence [82]. The role of these quinoline and quinolone derivatives as fluorescent markers, optical brighteners, luminophores, UV absorbers, and colorants for most biomolecules was determined [83].

### 2.19. Photochemical Applications of Imidazo[1,2-a]pyridine Derivatives

Some derivatives of imidazo[1,2-*a*]pyridine and their imidazo[1,2-*a*]pyridinium salts were used to prepare styryl dyes as these are well-known fluorescent compounds [84] and the function of peripheral benzodiazepine receptor (PBR) was performed by imidazopyridine-7-nitrofurazan conjugates known for their use as fluorescent probes [85]. The area of photochemistry involves a wide range of high technology applications of highly fluorescent heterocyclic compounds pyrido[2’,1´:2,3]imidazo[4,5-*b*]quinoline-12-yl cyanides as shown in Figure 10 [94].

### 2.20. Importance of Some Crown Ethers in Physical and Biochemistry

Several significant functions such as enhancement of crown ether–naphthalene derivatives by alkali metal ions and fluorescence quenching were reported by Sousa, but the reason for this fluorescence enhancement between 1,8-naphtho-21-crown-6(**15a**) and K^+^ (or Rb^+^) remained unidentified, with their structures shown in Figure 11. This fluorescence enhancement in dibenzo-18-crown-6(**15b**) in alcohol involved the chelation of alkali metal cations and depended upon their atomic number M^+^. In addition, higher temperatures (300 K) and smaller viscosity were also found to be responsible for fluorescence enhancement. The main reason for this enhancement was unknown, but experiments showed that the formation of planar or semi-planar structures in these types of complexes with a large ring size was comparatively harder to accomplish [86]. Physical chemistry and biochemistry are two major fields in which these studies hold very important places.

### 2.21. Application of Benzothiazole Crown Ethers as Metal Ion Sensors

Biological and environmental-related cations were considered hard to detect, but this was made possible by the most sensitive technique of fluorescence chemosensors. A few examples of such chemosensors or fluorescence-based metal ion sensors included benzothiazole fluorophore crown ethers, as shown in Figure 12 (**16a**–**c**). Apart from this metal ion sensing activity, the presence of nitrogen of the benzothiazole component provided extra binding capacity and so was considered of great interest as benzothiazole moiety was placed at ortho positions with respect to the crown ether. There were also chances of electrostatic interaction through ion–dipole interaction between the nitrogen ligand of benzothiazole moiety and alkali metal ions [87]. If salt concentrations were higher, quenching effects could be experienced and all these factors could lead to initial fluorescent enhancement.

### 2.22. Use of Fluorescent 14-Crown-4 Derivatives in Lithium-Ion Extraction

Lithium-ion extraction was attainable by some vital chromogenic 14-crown-4 (1,4,8,11-tetraoxacyclotetradecane) derivatives including crown *p*-nitrophenol **17a** and crown *p*-aminophenol **17b** type chromophores shown below in Figure 13. Extraction, fluorimetry, and spectrophotometry of lithium was achieved by a new fluorescent derivative 14-crown-4, 6-dodecyl-6-[2-hydroxy-5-(l,7-naphthalenedi-carboximido)benzyl]-1,4,8,11-tetraoxacyclotetradecane **17c**, which contained a *p*-(1,8-naphthalenedicarboximido) phenol moiety. Thus, the lithium-ion was selectively determined for extraction, fluorimetry, and spectrophotometry by this new 14-crown-4 derivative [88].

### 2.23. Biological Activity of Naphthoquinones

Some naphthoquinones show antiplasmodial and trypanocidal activities. These were tested by the cyclo-voltammetric activities of naphthoquinones. Many of the naphthoquinones show anticancer, anti-protozoan, and antibacterial activities (Table 2). In chemotherapy, they are the second most widely used heterocycles. Intercalation of bioactive oxygen in DNA double helix via reduction shows anticancer properties [89].

### 2.24. Nitro-Heterocycle and Their Biological Activity

Nitro-heterocyclic medications have long been employed as antibacterial, antifungal, and anticancer agents. Newer hypoxic tumor variants have received considerable interest. To be fatal, these drugs must decrease nitro groups, which is difficult in well-oxygenated cells. Hypoxia or anoxia makes them more poisonous and ineffective. In these conditions, they are more harmful and less effective. The electrochemical behavior of several nitro compounds was studied and compared. Some of the drugs selected include misonidazole, metronidazole, ornidazole, nitropyrazole, nitrofuran, and three nitrobenzenoid compounds. Their structures and reduction potentials vary, influencing their biological function [90].

### 2.25. Biological Activity of Imidazothiazoles

People who study the isosteric-related heterocycles, such as pyrrolothiazoles, imidazothiadiazoles, and imidazotriazoles, might want to look at how they work to treat different diseases, i.e., imidazothiazole has anti-psychotic, antifungal, anti-tumor, and anti-microbial properties, as well [91] (Table 2).

### 2.26. NHC (N-Heterocyclic Complexes) with Transition Metals

The *N*-based heterocycles form complexes with silver and copper metals, showing antibacterial, anticancer, antifungal, and antimicrobial activities. Their XRD shows the structure as shown in Figure 14 [92].

## 3. Conclusions

A large number of biologically important compounds contain the necessary conjugated double bond systems and are, therefore, potentially fluorescent. These include crown ether and *N*-containing heterocyclic compounds. Phosphorescence and fluorescence heterocyclic compounds, sometimes referred to as luminous materials, have received considerable attention because of their potential in various functional applications in organic electronics and/or optoelectronics and as materials of interest in pharmacology. These have various applications in the medicinal field as antioxidant, antimalarial, antitumor, anti-microbial, and antifungal agents. Quinolines have attracted the attention of scientists because of their uses in high-tech applications. Azafluoranthenes heterocyclic isomers may be explored as innovative, effective dyes for luminous or electroluminescent applications. Pyrene and its derivatives are often used as fluorescent probes in micellar systems for determining micro polarity, microviscosity, and aggregation number. More recently, heterocyclic azo compounds such as benzothiazole, pyrazole, and thiazole have been employed in electrochemical applications, biological applications, nonlinear optics, and structure–activity relationships [SAR]. The employment of fluorophores, including organic chromophores and crown ethers, which have high selectivity, sensitivity, and stability constants, opens up new avenues for research.

## Figures and Tables

**Figure 1 molecules-27-06631-f001:** Structures of azo dye compounds (**1a**–**1d**) showing anti-mycobacterial activity.

**Figure 2 molecules-27-06631-f002:** Structures of compounds (**2a**–**2c**) having antioxidant activities.

**Figure 3 molecules-27-06631-f003:** Compounds (**3** and **4**) with anti-hypertensive and antibacterial activities.

**Figure 4 molecules-27-06631-f004:** Structures of metal complexes of *N*-heterocyclic carbene compounds (**5**–**7**).

**Figure 5 molecules-27-06631-f005:** Structure of carprofen with anti-inflammatory activities.

**Figure 6 molecules-27-06631-f006:** Structures of compounds (**8a**–**8e** and **9a**–**9e**) showing anti-microbial activity.

**Figure 7 molecules-27-06631-f007:** Structures of compounds (**10a**–**b** and **11a**–**c**).

**Figure 8 molecules-27-06631-f008:** Structure of compounds (**12a**–**f**).

**Figure 9 molecules-27-06631-f009:** Structures of compounds (**13a**–**b**).

**Figure 10 molecules-27-06631-f010:** Structures of compounds (**14a**–**14e**) showing photochemical activity.

**Figure 11 molecules-27-06631-f011:** Structures of dibenzo-18-crown-6 (DC) and 1,8-naphtho-21-crown-6 ether (**15a**–**15b**).

**Figure 12 molecules-27-06631-f012:** Structures of benzothiazole crown ethers (**16a**–**16c**).

**Figure 13 molecules-27-06631-f013:** Structures of 14-crown-4 derivatives **17**(**a**–**c**).

**Figure 14 molecules-27-06631-f014:** Bis (1-allyl-3-butyl-2,3-dihydro-1*H*-benzo[*d*]imidazole-2-yl) silver complex.

**Table 1 molecules-27-06631-t001:** Anticancer activities of azo compounds (**1a**–**d**).

		IC_50_ (μM)		
Compounds	HCT116	A549	Jurkat	K562
**1a**	34.65 ± 0.35	˃50	˃50	˃50
**1b**	˃50	˃50	˃50	˃50
**1c**	43.33 ± 0.14	˃50	˃50	˃50
**1d**	48.19 ± 0.31	˃50	˃50	˃50

**Table 2 molecules-27-06631-t002:** Heterocyclic compounds and their properties.

	Compounds	Biological Properties	References
**1**	Azo dye compounds(**1a**–**d**)	Anti-bacterial Anti-tuberculosisAnticancer	[5,46]
**2**	3-aminopyrroles derivativesThiophene compounds	Anti-inflammatory AntibacterialAnalgesicAntiviralAntipyretic Anti-convulsant	[47,50]
**3**	2-aminothiophene product	Bio-imaging	[51]
**4**	Metalloporphyrins	Photochemistry and spectroscopyPotentiometric sensorsAntiparasitic	[54,55,56]
**5**	Fluorescent heterocycles (**2a**–**c**)Aminoadamantane compounds	AntioxidantNeuroprotection	[60]
**6**	Dihydropyridines (**3** and **4**)	Antihypertensive agentsCardiovascular protection	[61,62,63]
**7**	Metal *N*-heterocyclic carbine complexes (**5**–**7**)	Anti-microbialAntifungalAnticancer	[64]
**8**	Oxadiazole derivatives	Photo-reactivityPeptidomimeticsBioisosteres	[67,68]
**9**	2-chloro-5-methylpyridine-3-olefin derivatives(**8a**–**e**), (**9a**–**e**)	PhotobiologyPhotochemistry	[69]
**10**	Naphtha furans and thiophenes	Antitumor	[70]
**11**	Halogenated β-carbolines	Antioxidant	[71]
**12**	Flavonoids	AntioxidantsReduced cholesterol absorption Improved blood flowPhotochemistry and Photostability	[72,73,74]
**13**	Phthalocyanines	Antitumor Photodynamic therapyDrug delivery systems	[75,76]
**14**	Coumarins and phenanthridines	Intrinsic fluorescence propertiesLight-sensitive propertiesDyes and DNA targeting agents	[77,78]
**15**	Quinolones, quinolines and their derivatives(**10a**–**b**), (**11a**–**b**), (**12a**–**b**), and (**13a**–**c**)	Fluorescence Chemosensing Fluorescent probes for bacterial and tumour cellsMetal ions detectionFluorescent markersOptical brightenersLuminophores UV absorbers Colourants	[79,80,81,82,83]
**16**	Imidazole pyridine derivatives (**14a**–**e**)	Fluorescence Fluorescent probes	[84,85]
**17**	Crown ethers (**15a**–**b**)	Fluorescence Fluorescence quenching and enhancement Chelation of alkali metal cations	[86]
**18**	Benzothiazole crown ethers (**16a**–**c**)	ChemosensorsFluorescence-based metal ion sensorsQuenching effectsFluorescent enhancement	[87]
**19**	14-crown-4 derivatives(**17a**–**c**)	Lithium-ion extraction	[88]
**20**	Naphthoquinones	AntiplasmodialTrypanocidal AnticancerAnti-protozoans Antibacterial activities	[89]
**21**	Nitro-heterocycle	AntibacterialAntifungalAnticancer	[90]
**22**	Imidazothiazoles	Anti-psychotic AntifungalAnti-tumourAnti-microbial	[91]
**23**	*N*-Heterocyclic Complexes	Antibacterial AnticancerAntifungal Antimicrobial	[92]

## Data Availability

Not applicable.

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
