# Peer review of "Fluorescent and Phosphorescent Nitrogen-Containing Heterocycles and Crown Ethers: Biological and Pharmaceutical Applications"

_molecules, 2022, doi:10.3390/molecules27196631_

Round 1

Reviewer 1 Report

The review by Faiz Ullah and coworkers deals with the different biological applications of luminescent nitrogen-containing heterocycles and crown-ethers. The review is exhaustive in terms of applications and relatively easy to follow. However, at some points the manuscript looks like more as a list of compounds and it lacks discussion. It would be appropriate to discuss more deeply the photochemical properties of the different molecules mentioned and the impact of their luminescent features on their biological properties, as well as the structure / activity relationship. The use of tables would be helpful for such discussions.

Many references (for example: 31, 34, 39. 61, 68, 71) should be properly cited. English language should be carefully checked for errors (cispaltin; Fig. 4 instead of Fig.5 page 7; was taken serious; another mild stone;…).

Author Response

Response to Reviewers

Reviewer 1

The review by Faiz Ullah and coworkers deals with the different biological applications of luminescent nitrogen-containing heterocycles and crown-ethers. The review is exhaustive in terms of applications and relatively easy to follow.

However, at some points the manuscript looks like more as a list of compounds and it lacks discussion. It would be appropriate to discuss more deeply the photochemical properties of the different molecules mentioned and the impact of their luminescent features on their biological properties, as well as the structure / activity relationship.

Ans: Dear reviewer, Thank you so much for your valuable suggestions. We have added more discussion in the manuscript. All the changes are mentioned in the manuscript.

The use of tables would be helpful for such discussions.

Ans: Thanks for the suggestions. We have added a new table in the manuscript summarizing the biological properties of the discussed compounds.

Many references (for example: 31, 34, 39. 61, 68, 71) should be properly cited.

Ans: Thanks for the corrections. We have updated the references properly now

English language should be carefully checked for errors (cispaltin; Fig. 4 instead of Fig.5 page 7; was taken serious; another mild stone;…).

Ans: Thanks for the suggestions. We have corrected the language throughout the manuscript

Reviewer 2 Report

Please check the comments and suggestion in the attached file.

Be careful with the format and figure sequence.

Author Response

Reviewer 2

  1. 1 In some cases you have antifungal instead anti-fungal, as well as antitumor and anti-tumor, please correct it

Ans: Thanks for the correction. We have incorporated into the whole manuscript

  1. 2 check the latin words, must be in italics in your text

Ans: Thanks for the correction. We have incorporated in the whole manuscript

  1. 3 the pharagraph can be moved to the top

Ans: Thanks for the suggestion. We have moved the paragraph.

  1. 4 in the text, 3 matches antoxidant various anti-oxidant

Ans: Thanks for the correction. We have incorporated in the whole manuscript

  1. 5 Different symptoms like respiratory issues, long-term coughs, and tuberculosis are

 treated by various plants in African and Asian countries.

Ans: Thanks for the correction. We have corrected the sentence

  1. 6 check your matches anticancer vs anti-cancer

Ans: Thanks for the correction. We have incorporated in the whole manuscript

  1. 7 The triplet excited state showed better E (trans) → Z (cis) isomerization compared to the singlet exited state.

Ans: Thanks for the correction. We have corrected the sentence

  1. 8 Fig. 7 corresponds to 10a-b and 11a-c compounds

Ans: Thanks for the correction. We have corrected the figure citation in the manuscript

  1. 9 Grammatical Error mentioned

Ans: Thanks for the mention. We have corrected the grammatical mistakes throughout the manuscript and have carefully checked again

Round 2

Reviewer 1 Report

The manuscript has been corrected both in style and grammar, but no almost discussion has been added. However, the addition of a table to summarize the properties of the compounds is valuable. There are still issues with the references, as 82 to 85 lack journal name. Also with ref. 34 (name of author is not Faculty).

Author Response

Dear Reviewer,

Thank you so much for your suggestions. We have revised and corrected all the references in the bibiliography section. 

Reviewer 2 Report

No one, thank you for the edited file.

Author Response

Dear reviewer, 

Thank you so much for your support and encouragment.